# Fine-Grained Pests Recognition Based on Truncated Probability Fusion Network via Internet of Things in Forestry and Agricultural Scenes

**Kai Ma [1], Ming-Jun Nie [2], Sen Lin [3,\*], Jianlei Kong [4,\*], Cheng-Cai Yang [4] and Jinhao Liu [1]**

1 College of Engineering, Beijing Forestry University, Beijing 100086, China; makai2015@bjfu.edu.cn (K.M.); liujinhao@bjfu.edu.cn (J.L.)
2 Business Growth Business-Unit, Jing Dong, Beijing 100176, China; niemingjun@jd.com
3 Beijing Research Center of Intelligent Equipment for Agriculture, Beijing 100097, China
4 School of Artificial Intelligence, Beijing Technology and Business University, Beijing 100048, China; 2030601023@st.btbu.edu.cn
\* Correspondence: linseng@nercita.org.cn (S.L.); kongjianlei@btbu.edu.cn (J.K.)

**Abstract:** Accurate identification of insect pests is the key to improve crop yield and ensure quality and safety. However, under the influence of environmental conditions, the same kind of pests show obvious differences in intraclass representation, while the different kinds of pests show slight similarities. The traditional methods have been difficult to deal with fine-grained identification of pests, and their practical deployment is low. In order to solve this problem, this paper uses a variety of equipment terminals in the agricultural Internet of Things to obtain a large number of pest images and proposes a fine-grained identification model of pests based on probability fusion network FPNT. This model designs a fine-grained feature extractor based on an optimized CSPNet backbone network, mining different levels of local feature expression that can distinguish subtle differences. After the integration of the NetVLAD aggregation layer, the gated probability fusion layer gives full play to the advantages of information complementarity and confidence coupling of multi-model fusion. The comparison test shows that the PFNT model has an average recognition accuracy of 93.18% for all kinds of pests, and its performance is better than other deep-learning methods, with the average processing time drop to 61 ms, which can meet the needs of fine-grained image recognition of pests in the Internet of Things in agricultural and forestry practice, and provide technical application reference for intelligent early warning and prevention of pests.

**Keywords:** insect pest recognition; fine-grained visual classification; deep-learning neural network; soft-VLAD aggregation; gated probability fusion

## 1. Introduction

Plant insect pests are one of the main reasons for reducing agricultural staple crops yields and reducing the quality of fruits cultivation. Due to the variety of plants, insect pests, complex environmental factors, and frequent outbreaks, the insect pests have seriously threatened the safety and sustainability of the complex crop and food production supply chain, which plagues agricultural and forestry workers around the world. According to the statistical results offered by Food and Agriculture Organization (FAO), over 40% of global food crops suffer losses due to pests and diseases every year, with economic losses of more than USD \$220 billion [1]. In recent years, serious locust disasters have broken out in Africa, the Middle East, and South Asia, causing millions of acres of farmland destructed and massive economic losses. Millions of people in these regions are seriously threatened with malnutrition and starvation. Therefore, developing technologies to accurately classify the categories of various insect pests is crucial for disastrous prevention [2,3]. The accurate identification of pest species is a matter of effectively monitoring, pre-warning,

and preventing large-scale disasters, which are enabled to improve the production, quality, and supply of various agricultural and forestry production, as well as improve the management efficiency of the entire industrial chain [4].

With the significant advancements of artificial intelligence and sensing technologies, including Internet of Things (IoT), unmanned aerial vehicles (UAVs), automatic robots, cloud computing analysis, and 5G communication, the "smart agriculture & forestry" has already become the modern development trend in recent years through the full chain of real-time information perception, quantitative decision-making, intelligent production control, and precise personality management [5]. In terms of insect pest identification, farmers and researchers have already applied various information and intelligent technologies to improve the working performance of complicated produce tasks in practical smart farming [6,7]. With high-quality image data of insect pests acquired by various surveillance cameras and vision sensors, many machine-learning methods, including local binary patterns [8], support vector machine [9], fuzzy set, BP neural network [10], etc. have been applied to discriminate insect pest varieties. However, those classical methods mainly rely heavily on complex statical analysis and designed feature engineering to gain a modest performance. This process usually requires lots of time-consuming and hand operations in a subjective manner to tweak plenty of parameters for reaching a modest level of identification accuracy. What is worse, existing methods are trained on the limited plant data sets collected in a controlled laboratory environment, which cannot deal with the practical applications of pest identification in a natural agricultural and forestry context.

In recent decades, deep-learning technology has made great development in visual applications, including image classification, object detection, and other video caption, which have been a very promising candidate for a practical and generalizability solution for various agricultural and forestry tasks [11,12]. Illuminated by the multilevel brain's perception structure of human vision, deep-learning neural networks design a computing model composed of multiple processing layers and nonlinear activation modules so that it can automatically learn the higher-dimensional representation from large-scale training samples at a more abstract and general level. Moreover, with the comprehensive guidance of optimization strategies and various learning tricks, deep-learning neural networks (DNN) could achieve better performance surpassing human recognition or traditional methods on different visual recognition aspects [13–15]. At present, several deep-learning models have been used in the image recognition of pest species, which obtained the better or even the best results in different forestry and agricultural scenes. For example, a DNN-based classification framework based on the convolutional neural networks (CNN) was implemented to recognize insect pest invasion situation on isolated plant leaf [16,17]. Several deep-learning neural networks, such as VGG16 [18], ResNet [19], and Inception [20], have also been applied to classify pest species and achieved considerable performance.

In order to guarantee the amount of training data for training complicated deep-learning models, many studies have collected a large number of public data sets of plant pest categories. Mohanty et al. [21] collected an image data set named the PlantVillage, which contains 14 crop types and 26 pest categories with over 50,000 images. Then, AlexNet [22] and GoogleNet models are employed to identify various classifies with an accuracy rate of 99.35%. Konstantinos, K. P [23] also collected 87,848 leaf pest pictures of different fruits and vegetables such as apples and potatoes and adjusted the fully connected pooling and multiple learning rate training of VGGNet, which effectively reduced the number of model parameters and improved the recognition rate up to 99.53%. Similarly, Wu G. et al. [24] collected over 75,200 images covering 102 types of crop insect pests and framed more than 19,000 photos to solve the problem of target detection. On this basis, feature extractors such as VGGNet and ResNet are combined with the detector modules such as SSD [25] and YOLOv3 [26] to effectively verifies the significant benefits of deep-learning technologies in insect pest classification as well as marking the area where the pests located and counting the number of them. Moreover, some studies have improved the pest recognition model in parameter lightweight and structure compression

operations to accommodate the real-time application requirements of automatic robots and IoT devices. Liu Y et al. [27] performed migrating two lightweight convolutional networks, MobileNet [28] and Inception V3 [29], to realize the pest identification of leaves. After comprehensively balancing factors such as recognition accuracy, computing speed, and network size, the optimal model is selected and transplanted to the Android system on mobile phones. Similarly, Artzai et al. [30] performed super-pixelization preprocessing and fully connected layer optimization on the ResNet50 network to improve the pest recognition performance of winter wheat in actual scenes. The recognition time of a single picture is within 2.8 s with an 87% recognition rate, which initially meets the application requirements.

Although the above studies provide a reference and feasibility for the use of supervised deep-learning neural networks to identify plant insect pests, the efficiency and accuracy for plant pest recognition need to be improved since existing deep-learning algorithms remain challenging in a natural environment. The main problem is that the identification process of pest species in complex practical scenes is a fine-grained visual classification (FGVC) problem. As one of brand-new hot research in computer science and intelligence technology, FGVC is mainly to identify image samples belonging to multiple sub-level classes via retrieving objects under a meta-level category, which is more difficult than simple coarse-grained identification of traditional image recognition [31]. With the remarkable breakthroughs of deep-learning techniques, FGVC has enjoyed a wide range of applications in both industry and research societies for several decades, such as birds, dogs, cars, or aircraft types. Nevertheless, it is still a challengeable assignment to realize fine-grained pest identification by using deep-learning models at high precision. In detail, there are many fine-grained difficulties in identifying insect pests in complex agricultural and forestry scenarios. Firstly, in an actual planting environment, multi-source cameras are applied to collect a large number of pest images, which usually leads to the intraspecific difference phenomenon. In this regard, the same meta-level category contains vast image samples with viewpoints, illumination, definition, positions in significantly different. This interference influence of data sources and environmental factors causes that the models are easy to misjudge different samples coming from the same meta-category into other categories. Secondly, there are growth period states of different insect pests, which leads to obvious differences in the characteristics of different stages of the same kind, and different pests show certain similarities at some times. What is worse, there is another interspecific similarity problem for coarse-grained DNN to identify insect pests, which is caused by the fact that the global appearance of different meta-level classes may be highly similar except for several critical local parts. Traditional coarse-grained models lack the effective ability to handle this identification puzzle.

Therefore, it is necessary to design a specific algorithm based on big data and diversity for fine-grained insect pest recognition to inference different agricultural and forestry scenes successively in the practical application. The algorithm can be transplanted to unmanned aerial vehicles, robots, smartphones, and other intelligent devices for use. Inspired by multi-granular feature aggregation and gated probability fusion [32], we proposed an effective fine-grained recognition algorithm on the basis of optimized Cross Stage Partial Network (CSPNet), which aims to extract multiple granulometric information only relying on image-level labels addressing to large-scale multiclass pest species. Then, a soft-VLAD aggregation module is added to exploit the distance distribution relationship among different granulometric features, which have absorbed the potential associated knowledge of different partial areas to evaluate intraclass variances and tolerate noise interference in high-dimensional space. Furthermore, each predicting probability of different sub-models is fused as the combined prediction results by allocating decision-level gates, which is the better prior knowledge to distinguish the interspecific similarity and intraspecific difference problems in pest image data sets. Finally, transplanting the trained model to an intelligent device for use, obtaining pictures through the intelligent device, and identifying pest categories through the model. The whole framework can realize end-to-end training under

the guidance of a comprehensive loss function. With equitable evaluations on the pest data set, experimental results show that our proposed method achieves better performance compared to other state-of-the-art models, and the algorithm can be well applied to smart devices, including unmanned aerial vehicles and smartphones. Meanwhile, our model is based on multi-species big data, so our model has greater advantages in dealing with diversity issues. The excellent robustness and usefulness make our algorithm more suitable for identifying slight interclass discrepancy and handling large intraclass variances when distinguishing the fine-grained insect pest categories in various smart agricultural and forestry tasks.

The rest of the paper is organized as follows: The experimental data set of insect pests is introduced in Section 2. The details of the proposed algorithm are explained in Section 3. Then, Section 4 presents contrastive experimental results and performance evaluation. Finally, Section 5 concludes the whole work with future research prospects.

## 2. Experimental Data Set

In order to ensure the performance of identifying insect pests in the actual environment, the image data used in this paper comes in two ways. The first way is that we investigated the public data set of insect pests, e.g., IP102 [24]. We re-select high-quality pictures from these data sets as a part of the final data set. Another part of the data in this paper relies on the Internet of Things monitoring system of agricultural and forestry pests. The collection of pest image data is obtained through different image acquisition cameras and equipment in IoTs, including unmanned aerial vehicles (UAVs), overhead surveillance cameras, picking equipment, pinking robots, and smartphones, which are shown in Figure 1. Afterward, these data are transmitted to the background cloud server through communication forms such as CAN/4G/WIFI. After data storage, data processing, and model learning training, the results of fine-grained identification of pests and prevention recommendations are returned in a timely manner, which can more accurately reflect the dynamic situation of the disaster.

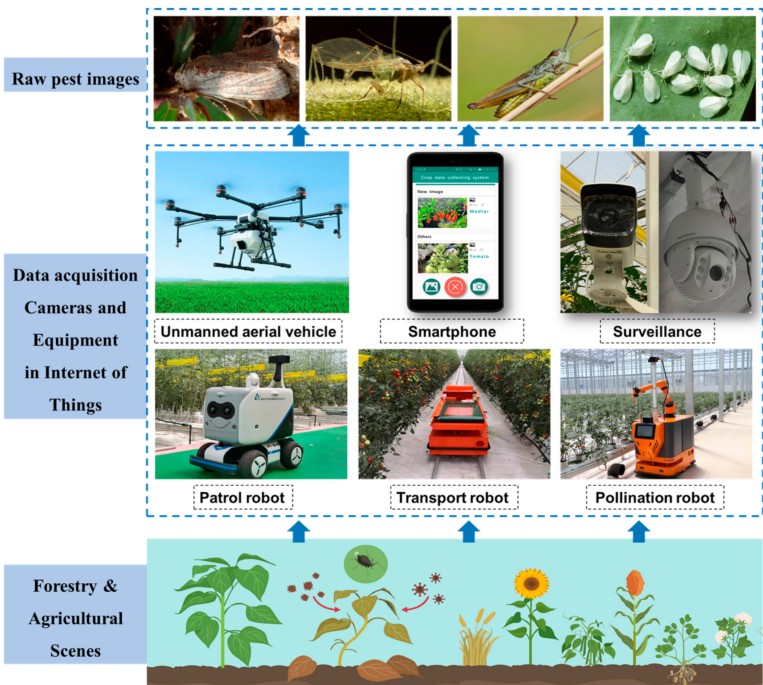

**Figure 1.** Image acquisition cameras and equipment.

The constructed image database and intelligence models in cloud servers will provide in-time identification and decision-making services for different equipment and devices based on their practical usage requirements. For example, in smart greenhouse manage-

ment, various cameras are connected in wireless and wired communication technology to form the Internet of Things monitoring system. Some surveillance cameras fixed on the roof and column will capture the large-scope situations of crops growth and pest distribution in different viewpoints. At the same time, the smartphone can quickly obtain the detailed health status of some specific plants or even close-up photos of a single pest. Other images from other automatic farming robots or UAVs can also be transferred into the cloud database, which calls for the online image procession and analysis to complete real-time agricultural operations. Thereby, those images taken by vast devices and cameras are different in resolution, size, focal length, etc. It is necessary to gather multi-source images to train the identification models for improving their generalization and extending applicable scenarios.

In view of the regional and seasonal distribution of different types of insect pests, the image collection of the pest samples in this article has been carried out in different periods from July 2018 to December 2020. A total of 74,437 effective pest images were collected in this work. Considering the different focal lengths, angles, and apertures of different camera equipment, the resolution of each photo was limited to 1000 × 1000 pixels. The constructed data set comes from leaves, flower, fruit, stem, and other plant parts of a total of 13 plant species (apple, cherry, corn, grape, citrus, peach, pepper, potato, strawberry, tomato, wolfberry, wheat, rice). Among them, a total of 132 insect pests come from 77 families, including Blattidae, Miridae, Cantharidae, Nolidae, Corduliidae, Limacodidae, Pterophoridae, Rhopalidae, Attevidae, Tettigoniidae, Crambidae, Spodoptera frugiperda, and so on, which are shown in Figure 2.

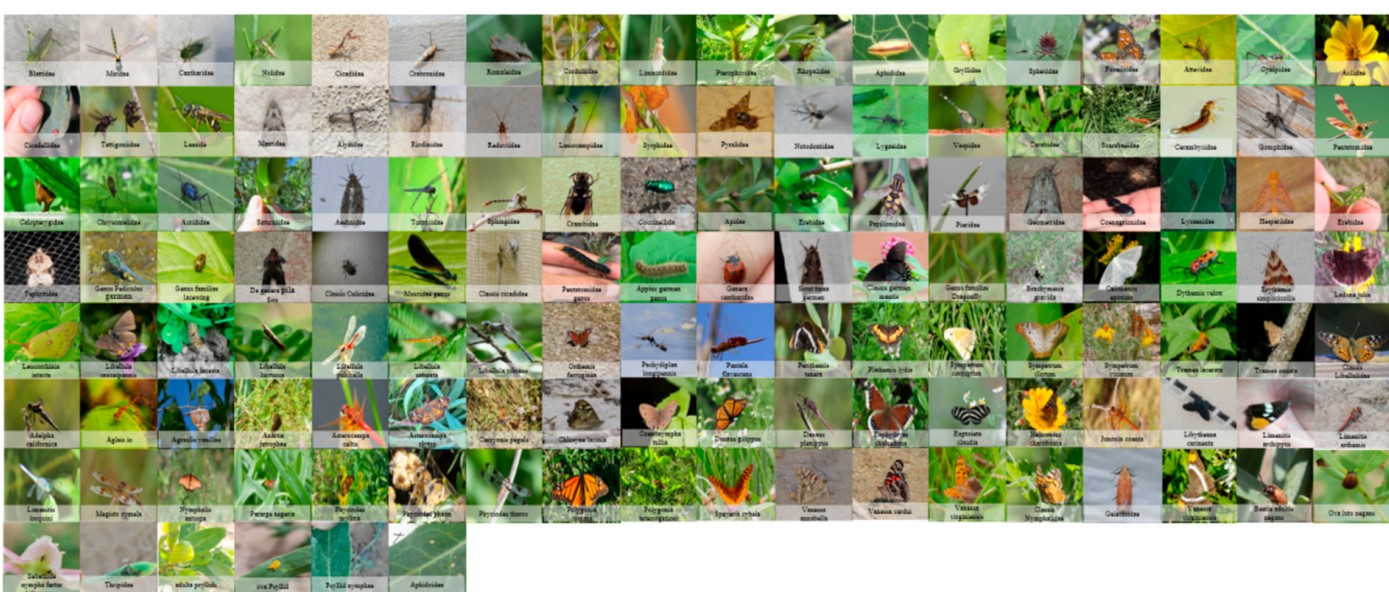

**Figure 2.** Examples of 132-class insect pest samples.

Considering the different occurrences of insect pests, the type distribution of various samples obtained in this paper is extremely uneven. The distribution of image data of each pest in the data set is shown in Figure 3. The columns are the number of each pest category, which has at least 100 image samples, which guarantees that all categories have the basic data amount that satisfies the model training. The largest number of samples is the 54th class Erebidae, with a total of 5109 valid images. The least number of samples is the 1st class Blattidae, with a total of 100 valid images. Therefore, the data set in this paper contains many fine-grained factors, which not only have similarities in different species and environments but also have significant differences in the periodic changes of the same species, plant organization, and light. Although it can describe the complexity of the real agricultural environment to a certain extent, it contains the recognition challenge of

fine-grained pests. This data set is a reliable experimental material to test the fine-grained classification of different models.

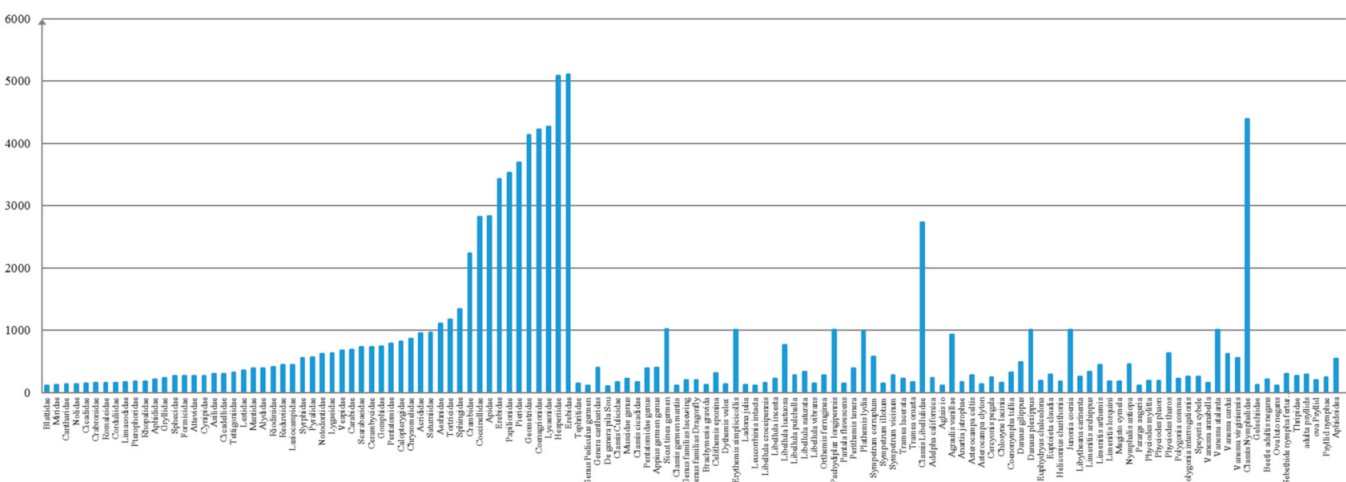

**Figure 3.** Sample number of different insect pests.

## 3. Materials and Methods

### 3.1. Data Augmentation

In order to compensate for the impact of uneven sample distribution on model recognition performance and to avoid overfitting the network, this paper performs enhancement processing on a small number of sample data before training. The enhancement method used in this paper mainly includes 5 steps:

(1) Random clipping: Randomly crop a rectangular region in the full photo with aspect ratio sampled in [4/3; 3/4], whose area is also randomly sampled in range of (10%, 95%). Then the cropped region will be resized into a 448-by-448 square image.

(2) Random rotation: Rotate part of the image randomly by $30°, 60°, 90°$ with a probability of 0.3 to simulate the view difference of the real landscape scene to improve the deformation adaptability.

(3) HSV Brightening: In the HSV color space, randomly varying the saturation of the images to (50% to 150%) of the original image. Then varying the brightness to (30% to 170%) of the original image, keeping the hue and contrast constant, to increase the light intensity variation and enhance the adaptability of the image.

(4) Gaussian blur: Add salt and pepper noise and Gaussian noise to the image, and simulate shooting samples with different sharpness to reduce image noise infection and level of detail.

(5) Mix-up: apply the effective way to obtain new training samples, which make use of linear overlay between any two images and their corresponding labels. The distribution is set at the value range of [0, 1]. The above enhanced steps can obtain the generalization of the network architecture and the improvement of the robustness capability of the model, and all images are randomly put for the network training after the above preprocessing.

### 3.2. Truncated Backbone Network

Since insect pests in the collected data set have not only many similarities among different meta-level categories but also are significantly different due to scenarios and sources discrepancy. When extracting the features of these images, the common deep neural networks usually only increase the magnitude degree of layers and functions with more complex structures to obtain better identification ability. However, this directly causes the model parameters to be too large, which is not suitable for the actual deployment of

IoT application. In addition, the model needs to make a balance between classification accuracy and small-scale parameters for preferable pest recognition.

Thereby, we introduce a compression strategy into a deep-learning method to design a truncated neural network, which could output the confidence recognition results in a lightweight parameter structure. The maintain operation is to replace the traditional convolution layers with multi-branch truncated modules consisted of small-size filters and efficient kernels, which doe not only increase the feature receptive fields but also reduce the number of parameter calculations. Taking the ResNet [32] as the basis, we designed the truncated Cross Stage Partial Network (CSPNet) as the backbone module to extract plentiful features. The optimized backbone can transfer multi-branch channel information into a higher-dimensional features combination, greatly improving identification accuracy as well as the reasoning speed of the whole method. In order to conduct the fine-grained crop classification tasks, after the pre-training process on the ImageNet data set, the truncated network can provide massive local and global features with different receptive fields and vector scales to perform fine-grained pests identifying. The detail of the proposed backbone mentioned above consists of three main components: the input process module, collateral CSP modules, and the output pooling module, which can be seen in Figure 4.

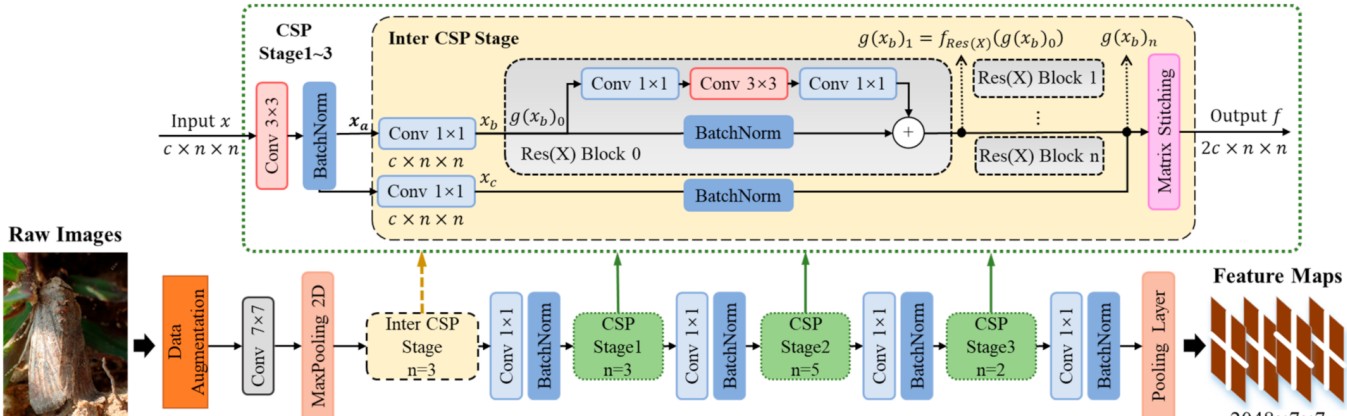

**Figure 4.** The architecture overview of the truncated backbone network.

In the original ResNet architecture, the inputs are usually set as default $224 \times 224$ images, limiting the information richness for training numerous network parameters. Instead, we use various data augmentations to adjust raw pest images into $448 \times 448$ pixel size. In order to train larger image data, the input layer is composed of a $7 \times 7$ convolution kernel with the step size set as 2. Another $2 \times 2$ matrix of maximum pooling with 2 step length is also added in the input layer, which outputs massive low-dimensional features with 64 channels for subsequent processing. Notably, the truncated CSP module is designed to passes through feature sets and explore their current internal laws, which is composed of an intermediate CSP stage and three continuous CSP stages. In the first inter CSP stage, 3 residual blocks are combined with partial BatchNorm and convolution layers to conduct various feature maps. Those intermediate features pass through the holistic network structure of inter CSP stage in some parallel paths. In detail, the maintain path goes through a $1 \times 1$ convolutional layer with 64-channels outputting, of which stride is set to 1 in the same zero padding. Then, the outputting vector will pass through some cascaded residual blocks such as ResNet or ResNeXt blocks. In each partial residual block, there are two tiny branches for extracting features at different scales. The first branch consists of three convolution layers with corresponding kernel sizes, respectively set as $1 \times 1$, $3 \times 3$, and $1 \times 1$. Then, the other branch is designed by using a BatchNorm layer to regular input value difference and alleviate the gradient disappearance. Subsequently, the results of two parallel branches are multiplied by each other to output the 128-channel feature vector. For the remaining residual blocks, the output can be acquired in the same

computational process to form the cascaded result of the whole maintain path. Similarly, the second path is designed by combining a $1 \times 1$ convolution layer and a BatchNorm layer with a 1-step stride and zero padding. As we directly splice the outputs of the maintain path and the second path by going through matrix stitching operation, we can obtain a middle output of 256 channels.

The remaining three CSP stages and the inter CSP stage has an obvious difference: by first passing through a down-sampling layer with $3 \times 3$ convolutions combined with a BatchNorm layer, we can obtain the middle output for other CSP stage 1~3, and the subsequent parts are the same as the inter CSP stage. Moreover, there is an additional interlayer consisted of $1 \times 1$ convolutional layer and BatchNorm layer between two adjacent CSP stages. The calculation processing of each CSP stage is shown as following:

$$
\begin{aligned}
x_a &= f_{3 \times 3}(x) \\
g(x_b)_0 &= x_b = f^b_{1 \times 1}(x_a) \\
g(x_b)_i &= f_{\text{Res}(X)}(g(x_b)_{i-1}) \\
x_c &= f^c_{1 \times 1}(x_a) \\
f &= (x_c \oplus g(x_b)_n)
\end{aligned}
\tag{1}
$$

where $x$ represents the input vector of each CSP stage. $f_{3 \times 3}()$ is the structure parameter of $3 \times 3$ down-sampling layer combined with BatchNorm layer, and the $x_a$ is the corresponding output. Similarly, $f^b_{1 \times 1}(), f^c_{1 \times 1}()$ are signifying the $1 \times 1$ convolutional operation going through two parallel branches severally. Then, $g(x_b)_0$ means the input vector of the 0-th residual block, which is equal to $x_b$. $g(x_b)_{i-1}$ means the input vector of the $i$-th residual block, which will pass through multiple branches to calculate the output $g(x_b)_i$. $f_{\text{Res}(X)}()$ represents the structure parameter of the partial residual operation. $x_c$ means the output vector of the relatively tiny path. $(A \oplus B)$ indicates the splicing operation between $A$ and $B$ in the channel dimension. The variable $n$ represents the number of residual blocks in different CSP stages, which is set as $n = \{3, 3, 5, 2\}$ for four stages. Finally, the global average pooling (GAP) is applied to establish the output layer of the whole truncated backbone, which exports the multi-dimensional feature maps $F$ with the channel number up to 2048. The expression is as follows:

$$
F = H_{csp}(\widetilde{x}, \widetilde{y})
\tag{2}
$$

where $H_{csp}()$ denotes the structure parameter of all cascaded layers. $(\widetilde{x}, \widetilde{y})$ is the input pest images and estimation labels. During the training and verifying process, the activation function is defined as Mish function, of which function expression is showed as:

$$
f(x) = x \cdot \tanh(\ln(1 + e^x))
\tag{3}
$$

*3.3. soft-VLAD Aggregation Module*

Through the difference analysis of massive feature maps for multiclass pest categories, it is well known that either using alone path's features or roughly integrating several paths' vectors is helpful enough for fine-grained pest identification. In fact, massive pivotal information is hidden in the complex network structure and various fitting operations, which can better characterize different features and their coupling association. Based on this, we construct an aggregation strategy through which the internal relationship of different granulometric features can be analyzed. The simple solution is to connect them with different scales and dimensions as a feature vector for better representation. However, it is hard to explore context characteristics as well as association among different feature maps, for the vector of local aggregated descriptors (VLAD) is a basic vector used to capture statistic information in low-dimensional space. Thus, the trainable soft-connect layer is proposed on the basis of VLAD to store the sum of residuals between all features and their corresponding cluster centers, then aggregates each local feature at the nearest cluster center, therefore fuse the different feature spaces into a unified meta-space. Using the outputting

features maps F as the input vectors, the soft-connect layer consists of a $1 \times 1$ convolutional operation and a k-mean clustering algorithm, which given multi-dimensional feature descriptors, computes a single D×K dimensional output to record the cluster centers $c_i$. The soft-connect output is written as $a_i$ according to the following equation:

$$a_i(x_l) = \frac{e^{w_i^T x_l + b_i}}{\sum_{i'} e^{w_{i'}^T x_l + b_{i'}}} \tag{4}$$

where $l$ corresponds to the dimension total of feature vector outputted by the multi-branch subnetworks, $i$ represents the serial number of cluster centers, and $i'$ represents the serial number of non-cluster centers. $w_i$, $b_i$, $w_{i'}$ and $b_{i'}$ represent the corresponding training stage updatable parameters, respectively. Then, after the soft-connect operation, a VLAD pooling layer is designed by combining intra-normalization and L2 normalization operations to unify the vector dimensionality. Finally, a full convolutional (FC) layer is added to complete the soft-VLAD module, which outputs a 256-dimensionality tensor $V$ as following:

$$V(j, i, l) = \sum_{j=1}^{N} a_i(x_l) \bullet \|x_l(j) - c_i(j)\| \tag{5}$$

With the soft-VLAD module aggregated, the first-order statistics of residuals between local feature $x_l$ and cluster $c_i$ in the descriptor space, representing different portions and slices of the object, are associated by the soft-connect weights, which achieves better flexibility and nonlinear expressive ability.

### 3.4. Gated Probability Fusion Module

Given the abundant feature maps on behalf of identification targets and their local components, a gated probability fusion module is designed to combinate feature information of multiple network branches and modules as the ultimate predicting results. This proposed module aims to form a universal network sub-module based on the gating theory, which will score different input features and dynamically configure the corresponding weights so that the classification results are optimized on a large scale.

Firstly, a gating network is proposed to analyze the complicated interactions of the input sub-modules through a self-gating mechanism in order to establish the nonlinear representation of probability associations among each network composition. With the internal associations, the gating network could eliminate redundant information between different feature maps and compensate for their interference effect on the final results. In detail, a concatenation layer was added after the soft-VLAD module to joint multi-dimensional feature maps from each sub-modules, which export an integral feature vector $f = concat(F_i, V)$. Take the feature maps $F_1$ and $F_2$ as an example, they are spliced with aggregated tensor V, respectively, to obtain the vector $f[E_1, E_2]$ as the input of the fusion module. Then, $f$ will go through the gating network to achieve the intrinsic weights and respective priorities of each composition by making corresponding output labels. With the non-sparse Softmax function, the vector equation $G_{ij}$ of the gating network outputs the $j$-th sample label of $i$-th feature maps as follows:

$$G_{ij} = Softmax(TopK(f \cdot W_g, k)) \tag{6}$$

where $W_g$ is a trainable matrix presenting the pertinence weights of all input sub-module and parameterizing calculations. $TopK_i$ means a sparsity ranking operation, which only captures the top $k$ gate values and sets the rest values to 0. Here, nonzero $k > 1$ is the gated parameter, which can be trained and tested in backward gradient propagation along with the whole method. *Softmax* represents the Softmax activation function. With sparse weighted concatenation, the gated network outputs a sparse $N \times n \times k$ dimensions tensor, of which $N$ is the number of sample labels and $n$ is the feature dimensions of sub-modules.

In the paper, $N$ is set up to 512 according to the batch-size value uniformly, and $n$ is set as 2048 equaling the output dimensions number of the concatenation layer.

Then, the next step is to calculate the predicting scores of each sub-module with gated weighting. This process is achieved by multiplying the distribution representation $E_{ij}$ of input features $f$ by the relevant gated vector $G_{ij}$, both of them are defined in the same dimension and matching scale. Then, the predicting scores belonging to different modules are calculated independently in Gaussian probability estimated solution as following

$$s_i = G_{ij} \odot E_{ij}(f) \tag{7}$$

$$E_{ij}(\widetilde{y}|\theta_i) = \frac{1}{\sqrt{2\pi}\sigma_i} \exp(-\frac{(\widetilde{y}-\mu_i)^2}{2\sigma_i^2}) \tag{8}$$

where $s_i$ means the predicting score of $i$-th sub-module. $E_{ij}$ is the probability estimation of Gaussian distribution. $\widetilde{y} = \{\widetilde{y}_1, \widetilde{y}_2, \cdots, \widetilde{y}_N\}$ is the input middle features corresponding to original pest images and annotations. $\theta_i = (\mu_i, \sigma_i^2)$ is estimated hyper-parameter according to normal Gaussian distribution, which contains the covariance matrix $\sigma_i^2$ and average value $\mu_i$. In addition, there is a normalized constraint of $\sum_{i=1}^{2} G_{ij} = 1$ to make sure the regularization of total probabilities. Then, the expectation maximization algorithm (EM algorithm) is selected to estimate the log-maximum likelihood of $\theta_i = (\mu_i, \sigma_i^2)$ in the fuse layer, which impends over the optimal result in constant iterations. Through local feature matching and internal correlation analysis, the gated fusion module could effectively concate various intraclass variations and distinguish small interclass differences for fine-grained pest identification.

In the last step, the cross-entropy(CE) loss function is required to estimate the inconsistency degree between the true annotations of each image and the relative predicting probabilities. The usual pathway of building loss function is to make some regularization constraints such as L1 or L2 terms in order to minimize the error separation between annotations and predictions to a small enough value. However, CE loss functions are dramatically encouraging adjacent predicting probabilities of different sub-modules similar to each other, which potentially causes the overfitting description of intraclass discrepancy. This easily leads to the very low interclass recognition in handling fine-grained pests with a similar appearance. Therefore, a novel fuse loss is proposed to synthesize multiple probability outputs of sub-modules to optimize overall model training. The equation of fuse loss is listed as follows:

$$FLoss = L_{fuse} + \lambda_1 L_{gate} + \lambda_2 L_{feature} = -\sum_{c=1}^{W} \widetilde{y}_c \log(s_c) - \lambda_1 \sum_{i=1}^{2} \sum_{j=1}^{N} G_{ij} TopK_i - \lambda_2 \sum_{i=1}^{2} \sum_{c=1}^{W} \widetilde{y}_c softmax(F_i) \tag{9}$$

The fuse loss is designed to cover three main sub-modules for identifying fine-grained pest categories, where $L_{fuse}$ means the united loss of soft-VLAD and fusion modules. $L_{gate}$ denotes the partial loss of the gating network, and $L_{feature}$ means the feature loss of the truncated backbone network. Then, two arguments $\lambda_1 \in [0, 0.5]$ and $\lambda_2 \in [0, 0.5]$ are introduced to balance the weightings of different compositions. $\widetilde{y}_c$ is the matching parameter, which is set to 1 if the predicting result of the $c$-th category is the same as true annotation, otherwise 0. $W$ indicates the number of pest categories. In the training and testing processes, the aforementioned fuse loss will guide the whole proposed method to approach the optimal solution, improving the overall performance for fine-grained pest identification. The detailed network structure of the above processes is shown in Figure 5 as follows:

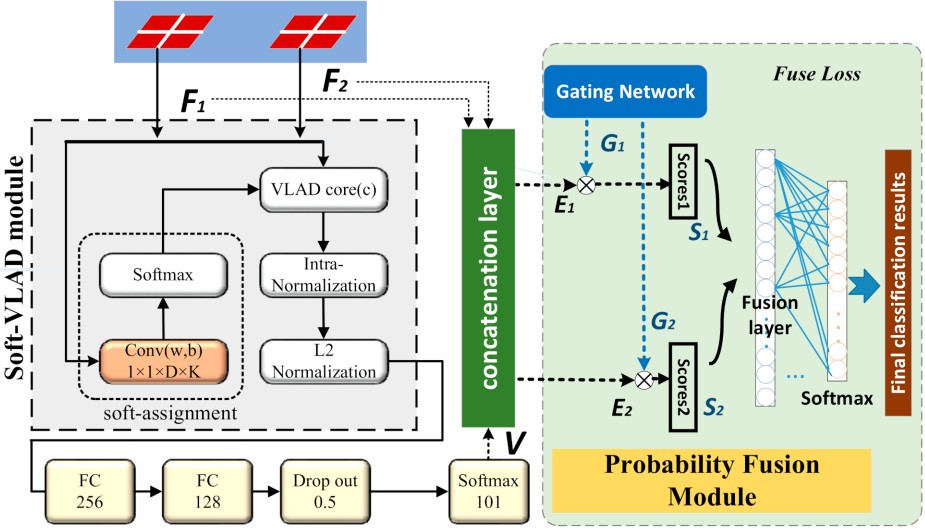

**Figure 5.** The cascaded architecture of soft-VLAD and gated fusion modules.

In this paper, some network hyperparameters are initialized to make sure the efficient training and credible learning of the entire model. The learning rate of the backbone network is set initially as 1e−3 in the first 30 cycles. Starting from the 31st cycle to the 140th cycle, the cosine annealing strategy is applied to reduce the learning rate in 70% attenuation ratio, until down to the minimum value 1e−7 of learning rate. The annealing step is set as 2, and each stage length is set for 20 cycles. During the training process, the size of the mini-batch was set to 128 according to the computing resources. The initial number of the gate value $k$ is set to 6. Then, the Adam optimization is selected as the default settings to dynamically turn the adaptive learning rate on or off and adjust massive parameters to optimal results. Moreover, we pre-trained the proposed network on the public ImageNet data set to achieve the pre-training weights, then loaded it to learn the fine-grained identification ability on professional pest data sets. This operation helps the whole training process approach to the preferable direction with faster convergence.

## 4. Results and Discussion

In order to make sure the dependability of network training, this paper randomly selects 15% of pest data sets as the testing images. The rest of the pictures are set as the training and verification data set. The proportion is 75% and 10%. Based on this division, this article uses Ubuntu 16.04LTS as the system to build a cloud server platform. This platform is equipped with a double-core processor with Intel Core i7-8600, a memory of 256 G, and a 4 × 4T solid-state hard drive. p40 GPU, computing cache is 96 G, using deep-learning framework Tensorflow1.4 and pytorch3.0.

In order to verify the performance advantage of the fusion model, this article trained multiple comparison models, including multiple coarse-grained deep networks such as Vgg19 [18], Resnet50 [19], Densenet121 [33], SEnet [34], and multiple fine-grained models such as NTS-Net [35], FAST-MPN-COV [36], and DFL [37]. The evaluation metrics in this paper includes: the top-1 and top-5 accuracies of measuring the overall performance of the model on the data set, using the F1-score (F1) to measure the overall robustness of the model, and ASP is used to indicate the average single sample recognition time. In order to verify the preferable ability of the proposed method for fine-grained pest identification, some practical experiments are accomplished compared with other classical CNNs method, which has already achieved remarkable success in other fields, to illustrate the better performance of the proposed algorithm. In other fields, those CNNs have obtained lots of remarkable success in many aspects. Additionally, some fine-grained models are also used to illustrate the effectiveness of the probability fusion module in our proposed method for

fine-grained pest classification. ASP, top-1 accuracy, and top-5 accuracy of each comparative model are listed in Table 1 as follows:

**Table 1.** Experimental results of ASP, top-1, and top-5 accuracies.

| Method | Top-1 Accuracy | Top-5 Accuracy | ASP (ms) | Parameter(M) |
|---|---|---|---|---|
| Vgg19 | 78.14% | 80.32% | 38 | 383.3 |
| SEnet | 79.09% | 82.25% | 46 | 376.5 |
| Resnet50 | 85.74% | 87.95% | 35 | 381.9 |
| Densenet121 | 81.38% | 84.14% | 33 | 396.2 |
| DFL | 75.81% | 79.83% | 74 | 254.3 |
| FAST-MPN-COV | 87.58% | 90.49% | 64 | 196.7 |
| NTS-Net | 87.12% | 89.57% | 57 | 120.3 |
| **FPNT** | **91.18%** | **94.85%** | **61** | **89.2** |

As shown in Table 1, It can be seen that compared to common coarse-grained models such as vgg19, Resnet50, etc., the fine-grained models obviously have more significant recognition accuracy, whether from top-1 accuracy or top-5 accuracy. However, from the perspective of the top-1 accuracy and top-5 accuracy of the single fine-grained model DFL, FAST-MPN-COV, NTS-Net in the table, although they have significantly improved compared to the coarse-grained model, their accuracy is also difficult to reach a satisfactory level. Through the corresponding comparison, we can see that after clustering at the NetVLAD layer, the corresponding fine-grained models such as FAST-MPN-COV and NTS-Net all have a little improvement in recognition accuracy, which shows that the feature clustering adopted in this paper has a positive effect. Secondly, from the results of the fusion of FAST-MPN-COV and NTS-Net, the accuracy of the fusion is significantly improved compared to the single model, which proves the effectiveness of the decision-level fusion strategy. The effectiveness of the above-mentioned NetVLAD clustering and fusion strategy is shown in the table. The proposed FPNT with a gated fusion strategy and NetVLAD modules achieves the best recognition effect. Its top-1 accuracy and top-5 accuracy reached 91.18% and 94.85%, respectively, which achieves the best recognition effect on this data set. Similarly, our end-to-end method consistently shows better performance on ASP outperforming state-of-the-art methods and has relatively compact multi-layer network architecture without extra annotation. This clearly indicates that FPNT has better learning and representation capabilities for actual pest recognition applications. In addition, we also compare the parameters of the models. From Table 1, we can see that the parameters of our model are the smallest, so our model is the best in terms of lightweight, which is more suitable for use on smart devices than other models.

Figure 6 provides more experimental analysis of precision-recall histograms of different categories, which can further illustrate the feasibility and effectiveness of FPNT. In comparison to other models, the proposed FPNT method achieves better performance on the average recall and precision indicators, of which value is, respectively, up to 94.89% and 93.25%. The results show that the FPNT model can excavate the advanced features with excellent presentation skills from large-scale pest images, which is a further response to the better stability and accuracy than other models. Thus, it is much suitable for pest identification tasks in agriculture and forestry applications.

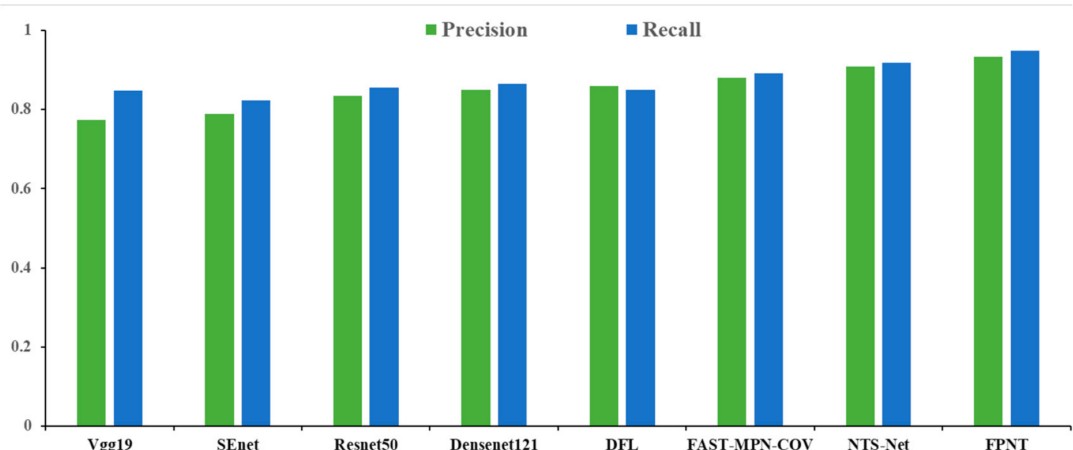

**Figure 6.** Precision and recall results of the comparison experiments.

In order to test the convergence effect of the model, we conducted an ablation experiment to compare the numerical changes of loss values offered by different methods through the control variables. The loss value of each model is estimated by calculating the inconsistency extent between all sample labels and the homologous prediction results, except for our method, which is achieved by using the fuse loss function as Equation (9). As shown in Figure 7, there are eight curves of loss value depicted in this experiment. From the overall evaluation, the smaller the loss value of one curve, the better the robustness of the relevant method. It is observed that the overall direction of loss curves offered by all comparative models has a downward trend, which shows that they all tend to converge and stabilize after 15 update epochs. Moreover, some popular coarse-grained models, including SEnet, Vgg19, Resnet 50, and Densenet-121, are obviously not as robust and effective as fine-grained classifiers during network training. This directly indicates that the pest image data set collected in this work has the cogent characteristics of fine-grained classification problems, and it is still challenging to existing deep-learning models without fine-grained techniques. On this basis, the proposed FPNT method has achieved the preferable performance in terms of loss value and curve changing than all models, which verifies the gated fusion and Soft-VALD aggregation modules work well in promoting the training efficiency and convergence speed. To a certain extent, the changes result of loss value also shows that the proposed FPNT method has a better classification ability than other models to handle the massive fine-grained pest images, furthermore suitable for various practical applications in complex environmental scenes.

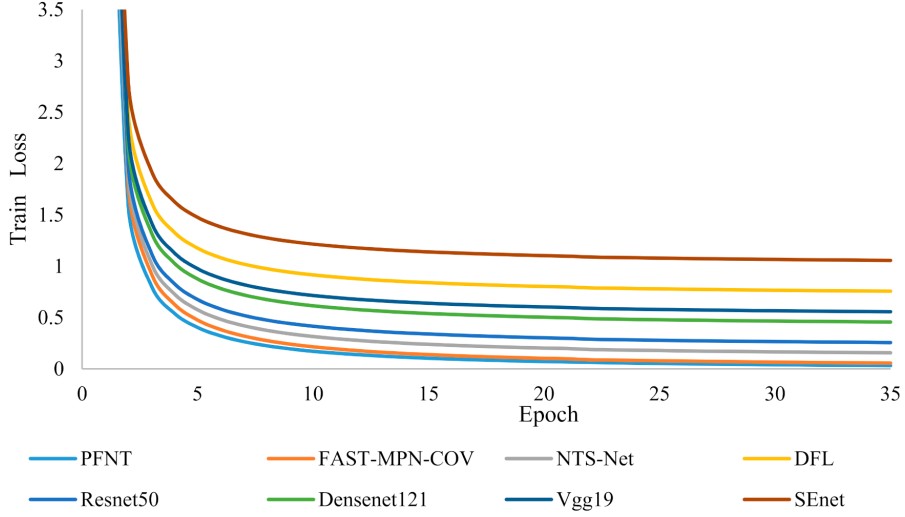

**Figure 7.** Comparative loss value curves of different models.

In order to illustrate the role of the fusion module, this paper carries out a visual comparison experiment by comparing the FPNT method with three fine-grained models. We analyze the accuracy of each model for each type of image on the data set, as shown in Figure 8. It can be found that a single model has different recognition capabilities for different categories and even achieves obvious accuracy discrepancy among each pest category. In fact, the detailed accuracy of each model for different samples in the same pest category is also diverse. With the help of a designed fusion module, FPNT can effectively integrate multi-dimensional features extracted by different modules and eliminate redundant information among various complements, thereby improving the recognition accuracy for each type of pest image. For example, Resnet50 has an accuracy of 63.6% in category 0 and 69.2% in category 3; Fast-MPN-COV has an accuracy of 90.9% in category 0 and 84.6% in category 3; after the gated fusing operation, the average accuracy of PFNT method in this 0-th pest class is up to 100%; the accuracy in the 3-th category was 100%. It can be found that the fused module makes the identification difference of diverse individual modules or methods for fine-grained targets gradually reduced, thereby improving the overall accuracy.

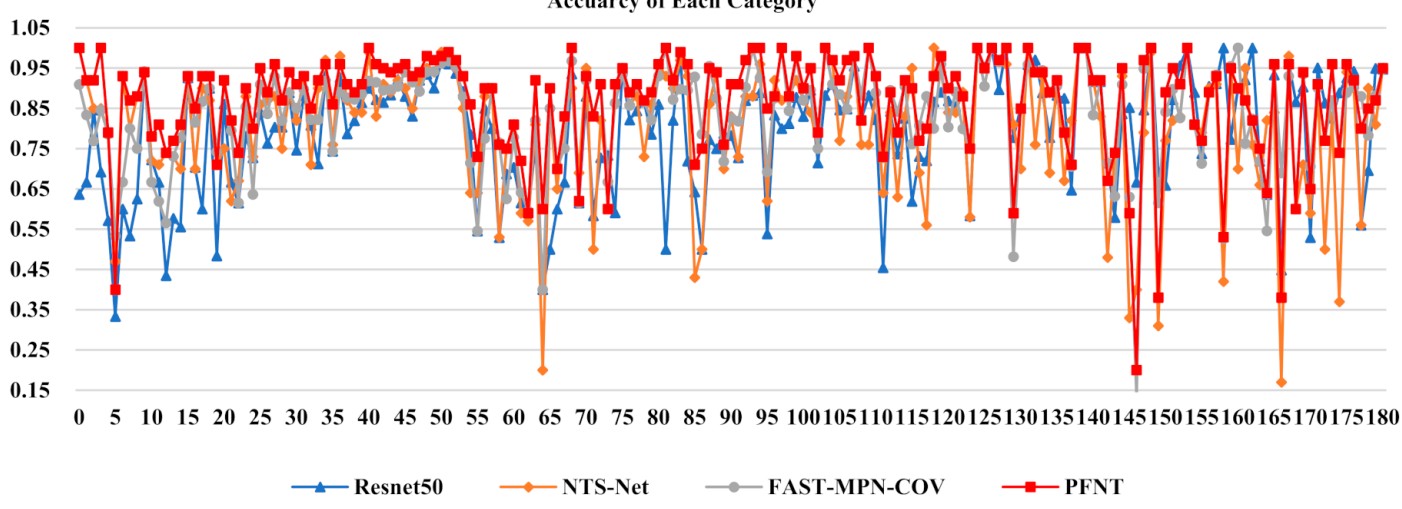

**Figure 8.** Accuracy of each type of the Resnet50, NTS-net, Fast-MPN-COV, and PFNT.

In addition, take some test samples (Geometridae of class 49, Classis Nymphalidae of class 120) as examples. This paper compares the recognition results under the four models of ResNet50, NTS-net, Fast-MPN-COV, and the fusion model. As shown in Figures 9 and 10, the red boxes represent the images that the model has misidentified, and the rest represent the images that are correctly identified. It can be seen that compared with other models and a single fine-grained model, the accuracy of the model recognition after fusion is greatly improved, and the reliability of the above results is also verified on the side.

In a word, the results of the method indicate that the model has the advantages of high accuracy, less time of processing, fewer parameters of model, stable operation, and fast convergence, which means that the improved model of us can effectively solve the problem of fine-grained pest recognition. Meanwhile, the small amount of model parameters means that the model is lightweight, which makes the model run effectively on other devices. In practical applications, the model can be directly deployed in various devices (such as unmanned robots and drones) to improve the ability to automatically recognizing pests, or the image can be transmitted through the device (e.g., various IoT cameras and smartphones) to the cloud server, in which the model deployed to provide fast and accurate identification services. Even if farmers do not have professional biological knowledge of pests, they can accurately know the category and obtain the corresponding

management and prevention strategies, which will effectively guarantee the production and quality safety of the entire plant.

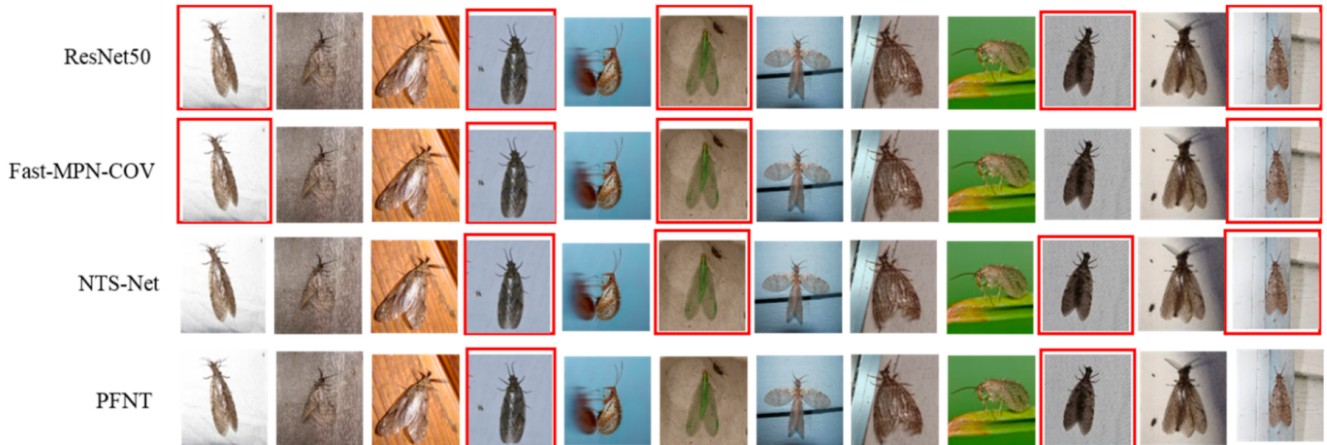

**Figure 9.** Comparing recognition results of 49th under each model.

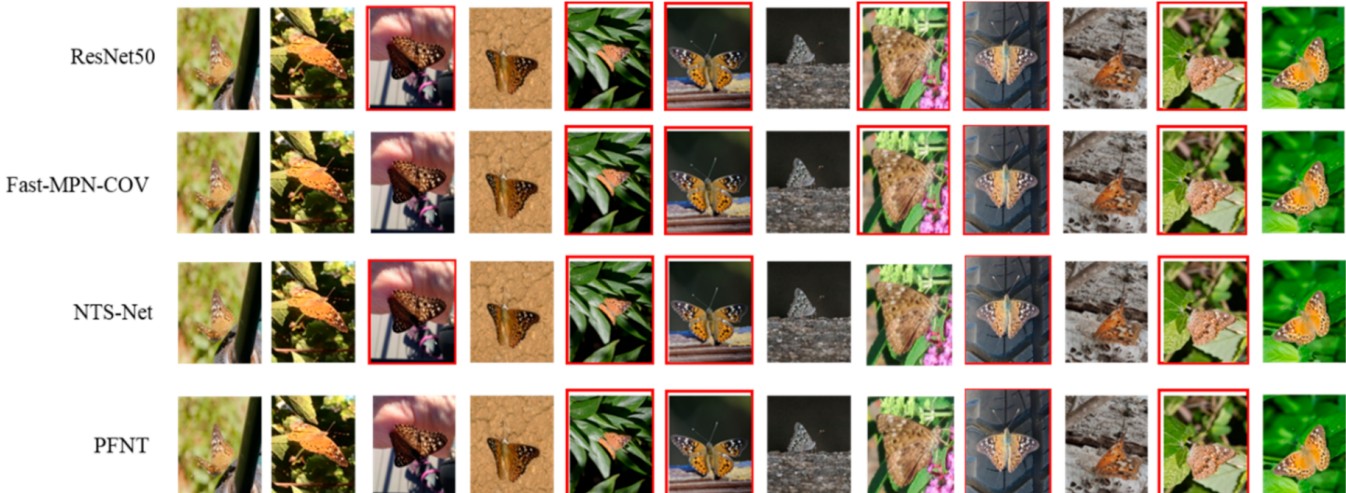

**Figure 10.** Comparing recognition results of 120th under each model.

## 5. Conclusions

Aiming at the complex situations such as pests that are widespread in agricultural and forestry applications, this paper uses a variety of IoT devices to collect images of 132 pests to construct a fine-grained data set suitable for practical operations. Based on this, this paper proposes an effective probability fusion network for the fine-grained identification of insect pests. After the data enhancement preprocessing, this paper uses the CSPNet network to design a truncation backbone structure as a fine-grained feature extractor. The model feature vector is input into the soft-VLAD structure for feature aggregation and high-dimensional representation. Finally, the confidence results of the sub-module are integrated into the gated probability fusion layer, and the maximum parameters are used to iteratively estimate the hyperparameters of the fusion layer to achieve end-to-end model training and fast convergence. The experimental analysis shows that the PFNT model in this paper has higher recognition performance, with accuracy as high as 93.18%, and its efficiency and robust performance are better than other comparative deep network methods. The average leaflet recognition time is 61 ms, which can meet the fine-grained identification of pests at the terminals of the Internet of Things in agricultural and forestry

practice and can play a positive role in scientific, reasonable, and effective early warning and control of plant pests for farmers.

In future work, we will attempt to put more pest identification networks for certain crops into practice. At the same time, we would like to apply more advanced algorithms to further optimize the parameters and improve the reliability and safety of the model. Moreover, it will be one of our key tasks to apply the model to more scenes.

**Author Contributions:** K.M.; M.-J.N. and J.K.: Conceptualization, Methodology, Writing—original draft. M.-J.N.: Formal analysis. C.-C.Y.; M.-J.N.: Investigation, Software. S.L.; M.-J.N.: Data curation, Writing—review and editing. J.L.: Supervision. All authors have read and agreed to the published version of the manuscript.

**Funding:** This research was financially supported by the National Natural Science Foundation of China (No. 62006008), National Key Research and Development Program of China (No.2020YFC1606801), Beijing Natural Science Foundation (No.4202014), Humanities and Social Sciences of Ministry of Education of China (No. 19YJC790028, No.20YJCZH229), and 2021 graduate research ability improvement program of Beijing Technology and Business University.

**Institutional Review Board Statement:** Not applicable.

**Informed Consent Statement:** Not applicable.

**Data Availability Statement:** Not applicable.

**Conflicts of Interest:** The authors declare no conflict of interest.

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
