# Peer review of "Fine-Grained Pests Recognition Based on Truncated Probability Fusion Network via Internet of Things in Forestry and Agricultural Scenes"

_algorithms, doi:10.3390/a14100290_

Round 1
Reviewer 1 Report
Dear authors,
Thanks for your paper. Paper is clear from scientific point of view and methods of analysis and results are clearly described. What is not clear for me is practical impacts of such technology. There are clearly described different repositories of pests. Different methods, how were this images collected were described. But what is missing, how trained algorithm can be applied in practical cases. Usually existing methods for collection of data are more suitable for monitoring of damages by pests, then identification of single pests Is intention, that there will be repository, where people will be able to send images of pests to identify this pests (I think, that usability is limited) or what is your idea. Please can you extend this in Discussion.
Reviewer 2 Report
The authors have done extensive work in trying to automatically identify pests with computer algorithms. While the modeling technique presented seems consistent, the paper falls short by greatly broadening the pest targets by not being able to give a detailed host description. Thus, we recommend an approach focusing on only one agricultural crop and its main pests for a modeling focused on results that can be useful for at least one agricultural crop. Another limitation was to use the discussion based only on the accuracy of the results, since we believe that the logic of the model used for classification is more important than the accuracy, considering that sources of error can occur from the phase of sample collection, image processing, among other sources. Thus, we encourage the authors to reformulate the paper with a focus on key pests from at least one host with more details about the crop evaluated.Author Response
Please see the attachment.

Round 2
Reviewer 1 Report
Dear authors,
thanks, your paper looks OK for me